# Re-Evaluation of Genotyping Methodologies in Cattle: The Proficiency of Imputation

**DOI:** 10.3390/genes14030547

**Published:** 2023-02-22

**Authors:** Moran Gershoni, Andrey Shirak, Yehoshav Ben-Meir, Ariel Shabtay, Miri Cohen-Zinder, Eyal Seroussi

**Affiliations:** 1Institute of Animal Science, Agricultural Research Organization (ARO), HaMaccabim Road, P.O. Box 15159, Rishon LeTsiyon 7528809, Israel; 2Beef Cattle Section, Newe-Ya’ar Research Center, Agricultural Research Organization, P.O. Box 1021, Ramat Yishay 30095, Israel

**Keywords:** genomic evaluation, genotyping platforms, single nucleotide polymorphism, *FABP4*, *ZHX2*, genomic imputation

## Abstract

In dairy cattle, identifying polymorphisms that contribute to complex economical traits such as residual feed intake (RFI) is challenging and demands accurate genotyping. In this study, we compared imputed genotypes (*n* = 192 cows) to those obtained using the TaqMan and high-resolution melting (HRM) methods (*n* = 114 cows), for mutations in the *FABP4* gene that had been suggested to have a large effect on RFI. Combining the whole genome sequence (*n* = 19 bulls) and the cows’ BovineHD BeadChip allowed imputing genotypes for these mutations that were verified by Sanger sequencing, whereas, an error rate of 11.6% and 10.7% were encountered for HRM and TaqMan, respectively. We show that this error rate seriously affected the linkage-disequilibrium analysis that supported this gene candidacy over other BTA14 gene candidates. Thus, imputation produced superior genotypes and should also be regarded as a method of choice to validate the reliability of the genotypes obtained by other methodologies that are prone to genotyping errors due to technical conditions. These results support the view that RFI is a complex trait and that searching for the causative sequence variation underlying cattle RFI should await the development of statistical methods suitable to handle additive and epistatic interactions.

## 1. Introduction

In dairy cattle, identifying polymorphisms that contribute to complex economical traits remains a challenge as gene effects fit an infinitesimal distribution, whereas, within breeds, the high levels of linkage disequilibrium (LD) often result in several significant effects being in complete LD [1,2]. This situation becomes even more demanding for traits that require complicated measurements such as residual feed intake (RFI, kgDM/day), which is defined as the difference between a cow’s actual dry-matter feed intake and its expected feed intake based on its size and growth. In dairy cattle, feed constitutes the majority (~60%) of farming expenditures, and improving feed efficiency has become of the utmost importance [3]. Inversely correlated with feed efficiency, RFI is a measurement of this trait. Although in lactating Holstein cows RFI is acknowledged to have a heritable component affected by multiple genes, to date, there is no polymorphism established as a causative factor and thus there is still a gap in knowledge of whether RFI could be reliably used to identify such a polymorphism [3]. Indeed, as demonstrated by Yao et al. [4], who applied the Random Forest approach as a machine-learning algorithm to the classification of this problem, numerous additive and epistatic interactions complicate the identification of causal variation [4].

Following quantitative-trait-loci (QTL) linkage-based studies, generally practiced as genome-wide association studies (GWAS), the identification of a causative polymorphism is achieved by targeted resequencing of small QTL intervals or by analysis of candidate mutations documented by whole-genome sequence (WGS) projects [2]. The causality of these candidates is examined by a concordance test [5]. This test compares the polymorphism zygosity states (homo- or heterozygous genotype) to the QTL segregation status for individuals for which this status had been determined using family-based analyses such as daughter and granddaughter designs [6]. Genotyping the candidate mutation requires efficient molecular methods that are suitable for low-scale non-multiplexed analyses, such as Sanger sequencing, SNaPshot, TaqMan, and high-resolution melting (HRM), of which the latter is considered the most cost-effective as it requires only a single inexpensive fluorescent dye [7].

Being a statistical methodology, genotype imputation predicts single nucleotide polymorphism (SNP) genotypes based on observed haplotypes in a reference population [8]. Using rules of Mendelian inheritance and basic principles of population genetics, models for genotype imputation rely on the fact that animals with common ancestors share haplotypes that extend over variable chromosomal distances [9]. If a reference population has WGS data and the related test population is genotyped for a SNP subset, it is likely that these populations have similar underlying patterns of LD [2,9] and, therefore, can be used as a cost-saving genomic strategy to infer the genotypes in the test sample [10]. Thoroughly reviewing this approach, Lashmar et al. [10] listed 14 imputation software programs commonly used in animal genetics research [10]. Of these, in the current work, we used Findhap.f90 and PLINK to compare and re-evaluate genotyping data of RFI studies indicating that BTA14 fatty-acid-binding protein 4 (*FABP4*) has a large effect on this trait [11,12]. We demonstrate the superiority of genotyping by imputation over the TaqMan, HRM, and even Sanger sequencing methods concluding that the relevance of the *FABP4* gene to RFI is smaller than has been previously estimated [11,12], compared to an additional gene candidate on BTA14.

## 2. Materials and Methods

### 2.1. Population and Phenotypes

A total of 192 Israeli Holstein multiparous dairy cows from the Agricultural Research Organization (ARO) dairy herd (Rishon-LeTsiyon, Israel) were used in the present study of which 114 cows were used in previous studies [12]. The RFI values were evaluated as previously described [12]. Briefly, the cows were healthy, multiparous, high yielding (35 kg/day), and 60–180 days in milking. Cows were held in one shaded corral as a single group in ARO’s experimental dairy farm, which is equipped with an individual feeding system. Computerized monitoring for each cow was performed for the number of meals per day (24 h), inspecting meal size and duration, eating over day and night, and daily feed intake. Cows had free access to water and, for six weeks, were individually fed ad libitum (5% orts) total mixed ration once daily between 9:00 and 10:00 a.m. The cows were milked three times per day.

### 2.2. Whole Genome Sequencing, Short Reads Mapping, Variant Calling, and Annotation

We followed a previously described WGS analysis pipeline [13]. Briefly, raw reads of 19 bulls (ENA BioProject PRJEB59761, [13]) were mapped to the reference genome (*Bos taurus* ARS-UCD1.2) [14]. Reads were aligned to this genome build by the Burrows-Wheeler Aligner software BWA-0.7.17 (bwa-mem algorithm, https://github.com/lh3/bwa, accessed on 13 February 2023). Next, polymerase-chain reaction (PCR) duplicates were removed using the Picard tool v2.20.2; (https://broadinstitute.github.io/picard/, accessed on 13 February 2023), and the curated BAM files were coordinated, sorted, and indexed using the Picard algorithm. Variant calling was carried out with the Genome Analysis Tool Kit v4.1.6.0 (GATK [15]) as recommended by the GATK workflow for germline short variant discovery. SNP and INDEL variants were called via local re-assembly of haplotypes by the HaplotypeCaller algorithm to generate an intermediate genomic variant call format (g.VCF) file. The g.VCF files were combined with the CombineGVCFs algorithm, and finally, a joint genotype call of the combined g.VCF file was carried out with the GenotypeGVCFs algorithm. All identified variants in the joint VCF file underwent comprehensive annotation by the Ensembl VEP [16].

### 2.3. Genotyping by Imputation

Blood samples were collected in EDTA vacutainers, and DNA was extracted with a GenElute Blood Genomic DNA kit (Sigma-Aldrich, St Louis, MO, USA) following the manufacturer’s protocol. A total of 192 cows were genotyped on the BovineHD BeadChip (777K SNPs), Illumina Inc., San Diego, CA, USA) by GeneSeek (Lincoln, NE, USA). Using the Linux awk command, genetic markers along BTA14 were selected in the analyzed windows (positions 16,732,918–16,848,253 and 44,656,120–44,690,793, see Appendix A). The dataset file was further manipulated by the addition of the targeted SNPs as markers with an indication of missing genotypes (ND in Dataset file) and the genotypes of bulls that were complete for all bulls based on the VCF file obtained following analysis of their WGS data (see Section 2.2). Following further formatting to fit the requirements of the imputation software, the imputation process was carried out using the findhap.f90 v4.0 [17] and PLINK v1.07 using the --hap-phase command-line option [18]. Findhap.options file included the following parameters: iters 6, Xchrom 0, maxlen 17, minlen 17, steps 1, maxhap 50, hapout 1, genout 2, damout 0, listout −1 and errate 0.00. For PLINK imputation the following command line was used: ‘p-link --noweb --tped file.tped --tfam file.tfam --cow --hap-window 17 --hap-phase’. For the analyzed dataset, haplotypes (best score = 1) indicated by PLINK were used for further analyses. Following imputation, the effects of the candidate alleles on RFI were assessed by linear regression, and coefficients of determination and regression were calculated using Excel (Microsoft Corporation, Redmond, WA, USA).

### 2.4. Validation by Sanger Sequencing

Sanger sequencing was applied to a limited sample, in cases of conflicting genotypes between the examined genotyping methodologies, or to validate the imputed genotypes. PCR was performed using primers designed with Primer3 [19] (Appendix A) and the Bio-X-ACT™ Long kit (Bioline Ltd., London, UK) according to the manufacturer’s instructions under the following conditions: 36 cycles for 30 s at 94 °C, 30 s at 59–63 °C and 30–50 s at 72 °C. Thereafter, the PCR products were separated on the basis of size in a 1–2% agarose gel stained with ethidium bromide. Following excision from the gel, the products were purified (Montage Gel Extraction, Millipore, Bedford, MA, USA) and sequenced using BigDye terminators kit 3.1 (Thermo Fisher Scientific, Waltham, MA, USA) run on an ABI3730 Automated Sequencer. Sequencing directions and primers are detailed in Appendix A.

### 2.5. Obtaining Candidate Variant around BTA14 RFI QTL

From the joint VCF file of 19 bulls with WGS data, all likely functional variants (i.e., missense, frameshift, stop gain/loss, and splice variants) and their genotype in each of the sires were obtained from +/− 1 Mb flanking the marker with the lowest probability of accepting the null hypothesis of no effect. A total of 18 likely functional SNPs were retained for further analysis. Assuming linkage disequilibrium between the BovineHD SNP associated with RFI and the hypothetical causative variant, both are expected to co-segregate. Therefore, the numeric representation of their genotypes (0, 1, 2) would correlate. The numeric representation was as follows: homozygote for the reference allele (0 non-reference alleles), the heterozygote (1 non-reference allele), and the non-reference homozygote (2 non-reference alleles). We thus calculated the Pearson correlation coefficient between the vector of the likely functional SNP genotypes, to the vector of the significant SNP marker genotype. These steps pointed out one SNP in the gene zinc-fingers and homeoboxes 2 (*ZHX2*) with a maximal Pearson correlation coefficient of 0.71.

## 3. Results

Equipped with an individual feeding system, the experimental dairy farm of ARO has been monitoring the RFI of lactating cows [12]. To further validate the findings of a QTL for RFI on BTA14 that have been based on a sample of 114 cows [12], we enlarged this sample to 192 cows all genotyped with the Illumina BovineHD BeadChip (777K SNPs). We also performed whole genome sequencing (WGS) for 19 Israeli Holstein sires, representing major lineages of the Israeli herd [20] with no RFI evaluations.

### 3.1. Imputing FABP4 SNP Genotypes

The indication that *FABP4* has a large effect on RFI mostly relies on the fourth and fifth SNPs (*FABP4_4* and *FABP4_5*) of 10 SNPs detected in gene *FABP4*, of which *FABP4_4* is capable of encoding a conservative substitution of hydrophobic amino acids [12] (Table 1). To infer unavailable genotypes for these SNPs in the enlarged test sample of cows and re-evaluate previous results, we examined an additional 15 SNPs within a short interval (~35 Kbp) of this gene locus (Table 2). This table is based on the VCF output obtained by GATK for 19 WGS projects of Israeli Holstein sires [20]. These sires served as the representative reference sample. In this bull sample, the high-quality scores and the high average of the depth of coverage recorded at the *FABP4* locus indicated segregation for all 17 examined SNPs and the accuracy of the determined genotypes (Table 2). Based on shared haplotypes between the test and the reference samples, alleles were imputed for all 17 SNPs of the combined population of 211 individuals. Capable of explaining 99.8% of allele genotypes, only four haplotypes were encountered, and none of them had a significant effect on RFI (Table 3).

### 3.2. Imputing ZHX2 SNP Genotypes

Following the indication (Section 3.1) that the effect of *FABP4* locus on RFI has been overestimated, we re-evaluated the association of BTA14 BovineHD SNPs with this trait using PLINK v1.07 and EMMAX (beta version) software (Appendix A). In these analyses, the significance of association peaked towards the centromeric end of the chromosome. Therefore, we investigated the presence of mutations that are likely to have a functional effect in this chromosomal region according to the reference sire sample. Excluding rare alleles, the joint VCF for these sires pointed to an SNP capable of encoding a conservative substitution of hydrophobic amino acids at the zinc-fingers-and-homeoboxes 2 gene (*ZHX2*, Table 1), that co-segregates with the top BTA14 SNP marker. To investigate the possible effect of this SNP on RFI by imputing its genotypes in the cow test sample, we examined an additional 15 SNPs within an interval (~115 Kbp) of this gene locus (Table 4). *ZHX2* locus showed segregation for most of the 16 examined SNPs in this bull sample (except for BTB-00553789, which showed no sequence variation among the 19 sires, Table 4). Based on shared haplotypes between the test and the reference samples, alleles were imputed for all 16 SNPs of the combined population of 211 individuals. To corroborate the imputed genotypes, we Sanger sequenced a sample of cows (*n* = 8) representing three possible zygosity states (Appendix A). Capable of explaining 98.1% of allele genotype combinations (sum of all haplotypes frequencies, Table 5), eight haplotypes were encountered, and one of them had an effect on RFI that remained significant even when applying a procedure of bootstrapping and accounting for multiple comparisons (Table 5). This haplotype was associated with *ZHX2* minor allele (T), which was also present in two other haplotypes that had no effect (Table 5).

### 3.3. Analyzing Genotyping Discrepancies

To investigate why the predicted effect of *FABP4* locus on RFI differs between studies, we compared the imputed genotypes to those previously recorded for the sample of 114 cows using the HRM and TaqMan methods [12]. Imputed genotypes were identical for the tightly linked *FABP4_4* and *FABP4_5* SNPs that are only 3387 bp apart. However, using HRM and TaqMan, as many as 25 cases (~22%) have previously displayed different genotypes for these SNPs in the same cows (Figure 1a). To evaluate the error rate of these genotyping methods, we Sanger sequenced both positions for all of these cases (exemplified in Figure 1b). In the resulting chromatograms, we observed additional variations (*FABP4_4′* and *FABP4_5′*, Table 1, Figure 1c) that helped to resolve noisy traces in which the chromatogram peaks of the alleles were imbalanced (illustrated in Appendix A). The Sanger sequencing results confirmed that *FABP4_4* and *FABP4_5* SNPs display the same genotype in all cases and that a substantial number of errors occurred during the HRM and TaqMan genotyping (Figure 1a). For two individuals (3187 and 3330), failed paternity verification using BeadChip data indicated that their BeadChip identification was erroneous, and these were removed from further analyses, reducing the size of the sample to 112 cows. Taking into account this size, error rates of 11.6% and 10.7% were encountered for the HRM and TaqMan methods, respectively, whereas imputed genotypes were all correct (Figure 1a).

To further investigate how these error rates affected results, we performed linear regression to estimate previously recorded *FABP4* SNP effects on RFI compared to the current data (Figure 2a–c). In Figure 2, the size of an SNP substitution effect and its ability to explain the phenotypic variance corresponds to the regression coefficient (parameter “a”, considering y = ax + b) and the coefficient of determination (R^2^), respectively. Thus, current results show a modest effect and ability to explain variance (Figure 2c) compared to the outdated data with a < −0.9 and R^2^ > 0.1, which is ~15 fold larger than the current (Figure 2a). Moreover, TaqMan genotyping errors were capable of skewing the effect size by reversing its direction, as evident from the positive slope in Figure 2b compared to the negative slopes in Figure 2a,b. The indication that *ZHX2* SNP (Figure 2d) has a larger effect on RFI and better explains it than *FABP4* SNPs further demonstrate that the effect of *FABP4* on RFI has been overestimated compared to other polymorphisms on BTA14.

## 4. Discussion

The availability of WGS and BeadChip techniques allows the comparison of accuracy between genotyping methods [20]. In this work, we re-evaluate genotyping methodologies that have previously been used to determine the effects of BTA14 SNPs on RFI [12]. As reviewed by Hayes and Daetwyler [2], in cattle, the combination of WGS and BeadChip information allows imputing whole-genome sequence data and has been proven to be a reliable tool for performing GWAS directly on the imputed sequence data. This capability has led to accurately identifying causative genes, if not causative mutations [2]. Affecting imputation accuracies in a tested sample, the size of the WGS reference population of the same ancestry and how much it represents the population’s genetic diversity should be taken into account. As few as 24 reference bulls have been used, indicating that estimated imputation accuracies were lower with a smaller reference population [21]. Here, we used only 19 bulls, and thus we carefully tested if our imputed genotypes matched those that had been previously determined for a sample of 114 cows from the same cowshed using the HRM and TaqMan methods for two tightly linked SNPs within the *FABP4* gene, one of which is capable of missense mutation [12]. Comparison between the genotypes of these SNPs had previously yielded moderate similarity (78% identical genotypes), yet, the imputed genotypes were identical, with over 10% mismatches to the previously recorded genotypes. To settle this discrepancy, for all cows that presented these genotype mismatches, we applied Sanger sequencing, which is still the gold standard method in determining the nucleotide sequence of DNA [22]. For all tested amplicons in this study, the accuracy of Sanger sequencing was further enhanced by genotyping additional variation that happened to occur within the same chromatogram near the target SNPs. This validation also allowed estimating copy-number proportions based on the chromatogram’s double peak ratios [23] and ruling out copy-number variation, concluding that uneven peaks (Appendix A) were likely the results of a secondary structure that may have promoted wobble-like pairing [24].

Sanger sequencing results indicated that the imputed genotypes were all correct and that few (*n* = 19) related individuals with WGS data were enough to yield accurate genotyping imputation, whereas, the HRM and TaqMan methods had an error rate of over 10%. Genotyping error rates as low as 3% and even 1% are thought to have serious effects on LD analyses [25]. Indeed, using the imputed genotypes, the effect of *FABP4* SNPs on RFI was as expected of the infinitesimal model for highly complex traits involving numerous additive and epistatic interactions such as the scenario that has been previously described for RFI [4]. To further demonstrate this and guided by PLINK and EMMAX analyses, we investigated the BTA14 centromeric to *FABP4* and were able to locate another missense mutation in the highly conserved homeobox gene *ZHX2*. In mice, *ZHX2* is a transcription factor with demonstrated centrality in liver metabolism, including lipid accumulation, enhanced inflammation, and hepatic fibrosis [26]. With a significant deleterious SIFT score of 0.03 (SIFT is an algorithm that provides a prediction of how likely nucleotide substitution is functional/damaging [27]), this mutation is likely to have a phenotypic effect; linear regression analysis indicated that it is more likely to associate with RFI compared to the *FABP4* missense mutation. Altogether, this suggests that attributing the effect of BTA14 on RFI to *FABP4* [11,12] was overtone. Moreover, considering genotyping errors and multicomparisons that accumulate by independent research groups, it is likely that setting the significance threshold to *p* ≤ 0.05 would result in random cases being overestimated as having large effects. Genotyping errors can be generated even with high-quality standards [25]. Our study demonstrates that practically, both HRM and TaqMan produce similar genotyping error rates, although these methods have been reported to differ in sensitivity to experimental conditions that may compromise them. For instance, HRM has large sensitivity of the melt curve shape to very small changes in environmental factors, including pH, ionic force, and cation concentration [28], and TaqMan is sensitive to the quality of the TaqMan probe [29].

## 5. Conclusions

Our case of a re-evaluation of genotyping methodologies used to study the RFI of dairy cattle demonstrated that genotyping by imputation based on WGS and BeadChip data produced superior results compared to other methods that are prone to genotyping errors due to technical conditions. Thus, genotyping by imputation should also be regarded as a method of choice to validate the reliability of the genotypes obtained by other methodologies. The results support the view that RFI is a complex trait and that searching for the causative sequence variation that underlies RFI should await the development of statistical methods that would be able to handle additive and epistatic interactions of more than two loci that influence parametric trait values.

## Figures and Tables

**Figure 1 genes-14-00547-f001:**
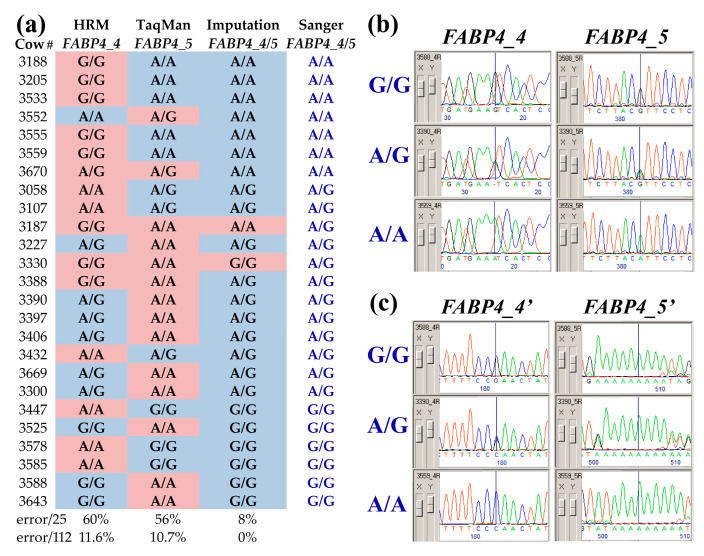
Resolving genotyping discrepancies in bovine fatty-acid-binding-protein 4 gene (*FABP4*). The imputed genotypes of two *FABP4* SNPs rs110757796 (*FABP4_4*) and rs133198078 (*FABP4_5*) were compared to those that have been previously described for a sample of 114 lactating Israeli Holstein cows, using HRM and TaqMan, respectively [12]. Imputation predicted identical genotypes for both SNPs (*FABP4_4/5)* and 25 discrepancies were observed with the previous results. This sub-sample of 25 cows was further validated by Sanger sequencing of PCR amplicons spanning both SNP sites (Table 1). (**a**) Background cell colors (blue—identical and red—discrepant) mark the genotypes obtained by the three genotyping methods compared to Sanger sequencing (*FABP4_4/5*, blue font). Error rates within the 25-cow sample and the 112 samples (excluding two misidentified cows) are indicated below. (**b**) Trace chromatograms obtained from three cows (3588, 3390, and 3559) were chosen to exemplify typical results at the SNP sites for the three genotype states (G/G, A/G, and A/A, blue font). (**c**) Typical trace signatures of additional variation near exon 2′s splicing site and in the gene promoter are also shown for other parts of the chromatograms of these three cows. These variations were tightly linked to *FABP4_4* and _*5*, respectively, helping to corroborate the three genotype states (blue font).

**Figure 2 genes-14-00547-f002:**
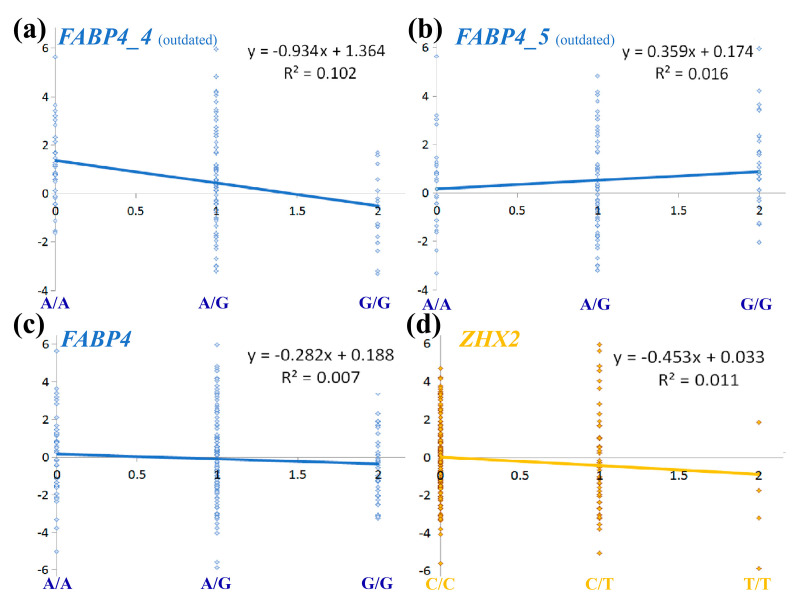
Effect of genotyping errors on analysis of the QTL for residual feed intake (RFI) on BTA14. Linear regression was used to draw the trendline showing the association between allele content (*x* axis) and RFI (*y* axis). Allele content was evaluated as a numerical value: 0, 1, 2 for major-allele homozygote, heterozygote, and minor-allele homozygote, respectively. For all *FABP4* analyses (blue), G was considered as the minor allele. Charts based on data previously published [12] are labeled as outdated. (**a**) Analysis of HRM genotypes for 113 cows of the *FABP4_4* SNP; (**b**) analysis of TaqMan genotypes for 113 cows of the *FABP4_5* SNP; (**c**) analysis of imputed genotypes for 190 cows of the *FABP4_4* or *_5* SNPs; (**d**) analysis of imputed genotypes for 190 cows of the *ZHX2* SNP (yellow). For this SNP, allele A was considered as the minor allele. In all charts, each dot denotes a single cow.

**Table 1 genes-14-00547-t001:** BTA14 SNPs.

SNP	Position ^1^	Alleles	Location	Effect	Minor	MAF ^2^
*ZHX2*	16,791,868	A/G	Exon 1	V163M	A	0.15
*ZHX2′*	16,791,746	A/G	Exon 1	sense	A	0.15
*FABP4_4*	44,677,959	A/G	Exon 2	I74V	G	0.5
*FABP4_4′*	44,678,114	C/G	Intron 1	ND	G	0.5
*FABP4_5*	44,681,346	A/G	Promoter	ND	G	0.5
*FABP4_5′*	44,681,230	Ins/Del	Promoter	ND	Del TAT	0.5

^1^ The ARS-UCD1.2 reference build, annotation release 106. ^2^ Minor allele frequencies were imputed for the 190-cow sample.

**Table 2 genes-14-00547-t002:** SNPs in BTA14 interval of the *FABP4* locus in a reference sample of 19 bull sires.

#	SNP	Position ^1^	REF ^2^	ALT	QUAL	AF ^3^	DP ^4^
1	BovineHD1400013246	44,656,120	G	A	12,288	0.60	27.2
2	BovineHD1400013247	44,656,664	C	A	12,469	0.57	27.4
3	BovineHD1400013248	44,659,016	T	C	13,561	0.64	30.1
4	BovineHD1400013249	44,660,779	G	T	15,796	0.64	27.6
5	BovineHD1400013250	44,662,403	G	A	13,575	0.64	25.4
6	BovineHD1400013251	44,664,380	A	C	13,808	0.63	27.5
7	BovineHD1400013252	44,665,702	A	C	14,162	0.64	32.9
8	BovineHD1400013253	44,666,155	A	G	12,912	0.62	31.0
9	BovineHD1400013254	44,666,683	A	G	13,615	0.62	39.1
10	BovineHD1400013255	44,668,063	G	A	13,459	0.60	35.2
11	BovineHD1400013257	44,676,672	G	A	13,689	0.60	29.0
12	*FABP4_4* (I74V)	44,677,959	T	C	11,719	0.67	27.3
13	BovineHD1400013258	44,680,255	T	C	12,639	0.63	27.3
14	*FABP4_5*	44,681,346	T	C	13,415	0.65	28.4
15	BTB-00567299	44,688,317	T	C	16,057	0.76	26.3
16	BovineHD1400013259	44,689,061	T	C	9444	0.38	34.6
17	BovineHD1400013260	44,690,793	A	C	3617	0.15	26.6

^1^ Allele position and orientation comply with the plus strand of ARS-UCD1.2 reference build. ^2^ REF, ALT, and QUAL follow the terminology of GATK’s VCF. ^3^ The allele frequency among the 19 sequenced sires. ^4^ The average depth of coverage in each position among 19 sequenced sires.

**Table 3 genes-14-00547-t003:** Imputed haplotypes in BTA14 interval of the *FABP4* locus and their effect on RFI.

Haplotype (SNPs 1-17) ^1^	Frequency	β ^2^	STAT	P
AACTACCGGAA**C**C**C**GTA	0.505	0.209	0.744	0.389
GCTGGAAAAGG**T**T**T**GCC	0.244	−0.388	2.120	0.147
GCTGGAAAAGG**T**T**T**ACA	0.232	0.066	0.055	0.816
GCTTGACAAGG**T**T**T**ACA	0.017	0.140	0.023	0.878

^1^ SNP order follows Table 2: bold-underscored positions 12 and 14 are of *FABP4_4* and *_5*, respectively. ^2^ Following the terminology of PLINK’s output for the --hap-linear option, β, STAT, and P are the regression coefficient, the coefficient t-statistic, and the asymptotic *p*-value for t-statistic, respectively.

**Table 4 genes-14-00547-t004:** SNPs in BTA14 interval of the *ZHX2* locus (~115 Kbp) in a reference sample of 19 bull sires.

#	SNP	Position ^1^	REF ^2^	ALT	QUAL
1	BovineHD1400005257	16,732,918	A	G	10,298
2	BovineHD1400005258	16,734,878	G	T	9563
3	BTB-00553789	16,736,646	A	G	NA ^3^
4	BovineHD1400005260	16,744,026	T	G	12,295
5	BovineHD1400005266	16,763,059	G	A	15,497
6	*ZHX2* (V163M)	16,791,868	C	T	3672
7	BovineHD1400005274	16,793,140	T	C	16,850
8	BovineHD1400005275	16,797,515	A	G	4502
9	BovineHD1400005278	16,805,570	T	G	7660
10	BovineHD1400005279	16,806,608	T	C	6181
11	BovineHD1400005280	16,808,205	T	G	7005
12	BovineHD1400005283	16,820,652	C	T	14,207
13	BovineHD1400005285	16,826,488	G	A	14,906
14	BovineHD1400005288	16,834,500	A	G	11,416
15	BovineHD1400005291	16,843,796	A	G	13,662
16	BovineHD1400005293	16,848,253	T	G	11,588

^1^ Allele position and orientation comply with the plus strand of ARS-UCD1.2 reference build. ^2^ Following GATK’s VCF format, REF, ALT, and QUAL are the reference or alternative allele, and the Phred-scaled quality score indicates the confidence in the variant called, respectively. ^3^ Not Applicable.

**Table 5 genes-14-00547-t005:** Imputed haplotypes in BTA14 interval of the *ZHX2* locus and their effect on RFI.

Haplotype (SNPs 1-16) ^1^	Frequency	β ^2^	STAT	P	EMP2
GTAGA**C**CATTTTAGGG	0.615	0.269	1.29	0.258	0.889
AGATA**C**CATTTTAGGG	0.103	0.351	0.902	0.343	0.954
AGGTA**C**TGGCGCAAAT	0.072	−0.801	3.37	0.068	0.414
GGATG**T**TGGCGCAAAT	0.062	−1.8	15.9	9.59 × 10^−5^	7.46 × 10^−4^
AGATA**T**TAGCGCGAAT	0.048	1.17	5.47	0.020	0.148
GGATA**T**TGTTTCGAGG	0.034	−0.346	0.296	0.587	0.998
AGGTA**C**CATTTTAGGG	0.029	1.16	3.12	0.079	0.464
AGATG**C**CGGCGCAAAT	0.019	−1.32	1.75	0.187	0.785

^1^ SNP order follows Table 4: bold-underscored position 6 is of the *ZHX2* (V163M) SNP. ^2^ Following the terminology of PLINK’s output for the --hap-linear and --mperm 1,000,000 options, β, STAT, P and EMP2 are the regression coefficient, the coefficient t-statistic, the asymptotic *p*-value for t-statistic, and the empirical *p*-value, respectively (after 1,000,000 bootstrapping and correction for multiple testing).

## Data Availability

Genotyping data directly involved in this study is available under Appendix A. Restrictions apply to the availability of additional genotyping data. Data were obtained from the Israel Cattle Breeding Association (ICBA) and are available from the authors with the permission of ICBA. WGS data is available from the European Nucleotide Archive (ENA BioProject PRJEB59761).

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
