# Peer review of "Re-Evaluation of Genotyping Methodologies in Cattle: The Proficiency of Imputation"

_genes, 2023, doi:10.3390/genes14030547_

Round 1
Reviewer 1 Report
In this study, authors have established the importance and accuracy of genotyping imputation for investigating complex trait genetics with practical applications concerning residual feed intake in dairy cows. Overall, the research design is well established and manuscript is of great quality. Here are a few minor issues outlined:
Line 44: Not sure if you can classify GWAS as a 'linkage analysis' method, as it is completely based on linkage disequilibrium mapping. I understand that the word 'linkage' might cover both, however terminologically this word is used for that of LOD score-based linkage mapping studies whereas GWAS is used for linkage disequilibrium mapping.
Line 66: Better to replace 'proficiency' with 'superiority'.
Line 68-69: Which gene? Specifying the gene name here is suggested.
Line 87: Replace 'Desember' with 'December'.
- Certain statements in various sub-headings of the Results section appear to be more like discussion sentences. Therefore, authors are suggested to revise the results and move those argumentative and explanatory sentences to the discussion section.
Author Response
Comments and Suggestions for Authors
In this study, authors have established the importance and accuracy of genotyping imputation for investigating complex trait genetics with practical applications concerning residual feed intake in dairy cows. Overall, the research design is well established and manuscript is of great quality. Here are a few minor issues outlined:
Line 44: Not sure if you can classify GWAS as a 'linkage analysis' method, as it is completely based on linkage disequilibrium mapping. I understand that the word 'linkage' might cover both, however terminologically this word is used for that of LOD score-based linkage mapping studies whereas GWAS is used for linkage disequilibrium mapping.
We are not the first to classify GWAS as a 'linkage analysis' method. E.g., as indicated in figure 5 of the following CellPress's review "GWAS is a linkage analysis that includes…" (Yugi et al., 2016. Trans-omics: how to reconstruct biochemical networks across multiple 'omic' layers. Trends Biotechnol. 34:276-290. https://doi.org/10.1016/j.tibtech.2015.12.013 ). Yet, to be more explicit, we modified “linkage analysis” to “linkage-based analysis”.
Line 66: Better to replace 'proficiency' with 'superiority'.
Done.
Line 68-69: Which gene? Specifying the gene name here is suggested.
Done.
Line 87: Replace 'Desember' with 'December'.
Updated to February 2023.
Certain statements in various sub-headings of the Results section appear to be more like discussion sentences. Therefore, authors are suggested to revise the results and move those argumentative and explanatory sentences to the discussion section.
Reviewer 2 Report
Line 34-37: Please provide a definition and/or explanation of RFI. Trait definition, unit of measurement, when/how it is measured, etc. You have to mention what it relates to i.e. feed efficiency. So, please elaborate. On a different note: if the focus of this application (i.e. imputation) is on/towards RFI – should the trait not form part of the title?
Line 46: Not only re-sequencing? What about SNP genotyping data? How is targeted re-sequencing more beneficial in comparison to SNP genotyping? Is low-density SNP genotyping not more affordable?
Line 50: Please elaborate on/explain what “zygosity states” means.
Line 65: Please “introduce” the FABP4 gene. Full name? Function? Locality? Has this gene been previously identified to play a role in RFI?
Line 84: What pipeline? I suggest “We followed a previously-described whole-genome sequencing (WGS) analysis pipeline (Ref)”
Line 91: Please be specific. What GATK pipeline? Aren’t there many?
Line 55/Line 91/etc.: Throughout, please check that all abbreviations are defined upon the first usage e.g. SNPs
Line 100: Please be specific. What commercial kit? And following the manufacturer’s protocol?
Line 101-102: This does not make sense to me. Is the 777K not an Illumina® genotyping panel? Was GeneSeek the service provider that performed the actual genotyping? Please be specific.
Line 102-103: I do not understand this step. Can you please rephrase and/or elaborate? Does it mean that you only extracted and utilized SNPs in the region on BTA14? If so, please specify the region (i.e. base pair range). **In the previous section (i.e. 2.2), you then also need to provide more details on the WGS data – before the pipeline was utilized, what was the targeted coverage, etc.? And on what platform was the WGS done?
Line 107-111: The imputation section needs to be stand-alone. It needs to be elaborated on – please specify the parameters used (even if the default was used), and the reference-versus-validation population used (if so), the underlying imputation methodology used within the software. I would suggest restructuring the entire M&M. It is not explicitly stated whether all data types were available for the same animals (i.e. did each animal have WGS, 777K, and Sanger data available?)… I would start off with a section on each data source i.e. WGS data > SNP genotyping panel data > Sanger sequencing data, and then imputation. Also, how was the imputation accuracy quantified? And what is the “RFI” column in the supplementary material? Are they phenotypes – there is no mention thereof in terms of origin (i.e. service provider), units (are they EBVs or performance records), etc., and needs to be added to the M&M?
Section 2.5 requires improvement in terms of detail. E.g. “Then, the segregation of the candidate alleles with the marker allele was examined.” This is very vague. There is no mention of software and/or the actual method/formula/train of thought that was used. Is this a verified way of doing this? E.g. “Since the population frequencies of the candidate alleles were lower…” – how were the frequencies calculated? Or were they given (if so, where)?
Line 132-137: This belongs in the M&M section – please see my comments for Lines 107-111 above. There is also no mention of/reference to Table 1? Are the SNPs in Table 1 known variants? This needs to be mentioned in the Introduction. Also, there are many vague column headings – what is “Minor”? As per your footnote, how was MAF calculated? This is not mentioned in the M&M…
Line 142-144: Not certain what this sentence means? This detail (e.g. about the gene of interest and the SNPs harbored within/around its genomic location) should be discussed in your introduction.
Line 151-152: What were the “test and the reference samples”? This needs to be discussed in your M&M. Are they related somehow? How do you know they share haplotypes? Furthermore, how do you know the imputation was accurate? This is not a very big sample size.
Line 168: Who indicated this? This is part of the motivation for this study, so it needs to be very clear.
Line 167-188: This section is very confusing. Some parts belong in the M&M, and others are discussions (not results). I would suggest a combined Results and Discussion… You also reference too many different tables – it gets confusing. Each table should be discussed separately.
Line 184: “Capable of explaining 98.1%...”. How was this calculated (no mention in M&M)?
Line 196-199: This analysis was never mentioned in the M&M? What software was used to do the significance testing? If this was mentioned in the M&M, no footnote would be necessary.
Line 201-207: This needs to be described in the M&M. It should not be necessary to explain how things were done in your Results section – only results should be reported.
Line 213: “For two individuals (3187 and 3330),…” Not necessary to mention animal IDs – the reader knows nothing about these animals – rather describe them in terms of sex or age.
Line 216: How were these error rates calculated?
Figure 1a): It should not be necessary to include this (i.e. an Excel spreadsheet indicating genotypes per individual) – provide means or proportions of animals with correct versus incorrect (or differing) genotypes.
Line 220: This is the first time the full name of the gene is mentioned…
Figure 1’s title is too long. The majority of this should move to the text.
Line 235-236: The regression analysis is never mentioned in the M&M.
Line 237: “In this figure,…” What figure? Please be specific.
Line 240-243: Please rephrase – this seems like an important result; however, it is not clear enough. I am also not sure what the second sentence is saying (the one referring to Figure 2b).
Line 269-270: Yes, but how was imputation accuracy determined? And what was the composition of the reference population in terms of relatedness etc.? Were these high-impact bulls? If so, you need to provide some more details as this is one of the most important aspects of imputation. If the haplotype diversity is sufficiently captured, and with WGS information nonetheless, then the size of the reference data set should have less of an effect.
Line 274: What is the 78%? What parameter? Is it a concordance rate? You need to be specific on what “similarity” means.
Line 286: How was relatedness quantified?
Line 292: This implies that you used EMMAx in this study, however, it is not mentioned once in your M&M? Also, what PLINK functionality was used? Also association analysis? Not clear at all.
Line 294-296: In what species?
Line 296: What is SIFT score? I am not following.
Line 303-309: I am not sure if I am missing something but there is a lot of focus on HRM and TaqMan methodologies but these are not included in this study. The focus should be on the Sanger, SNP genotyping, and WGS data.
There is not enough discussion on the relevance, impact, and possible benefits of your results. How can the results reported here be used to optimize future genotyping strategies? Is it a cost-benefit? The take-home message and purpose of the study are not coming across very well. How does this affect the feasibility of certain data types?
Line 311: “Our case of association study…” Is this an association study…? I think you need to change your title as it is misleading considering the results reported. My perception was that imputation methodology is the main focus, however, that is not the case.
Author Response
Comments and Suggestions for Authors
Line 34-37: Please provide a definition and/or explanation of RFI. Trait definition, unit of measurement, when/how it is measured, etc. You have to mention what it relates to i.e. feed efficiency. So, please elaborate. On a different note: if the focus of this application (i.e. imputation) is on/towards RFI – should the trait not form part of the title?
The focus of this manuscript is the proficiency of imputation. We use the data of the RFI project to demonstrate this, whereas, the findings related to RFI are of minor importance. Yet, we now better introduced this trait rephrasing: "… residual feed intake (RFI, kgDM / day), which is defined as the difference between a cow's actual dry-matter feed intake and its expected feed intake based on its size and growth."
Line 46: Not only re-sequencing? What about SNP genotyping data? How is targeted re-sequencing more beneficial in comparison to SNP genotyping? Is low-density SNP genotyping not more affordable?
Please note that this paragraph is not describing QTL mapping but rather "the identification of a causative polymorphism". Thus, SNP genotyping data is unlikely to help, except in the rare cases in which the SNP panel included the causative polymorphism. Nevertheless, in such case this would fall under "mutations documented by whole-genome sequence (WGS) projects", which are generally used also for establishing SNP panels.
Line 50: Please elaborate on/explain what “zygosity states” means.
For a certain heterozygous SNP in the parent, if you divide the progeny into two groups by the parent SNP alleles and detect significant difference between the groups' phenotypes than you may conclude that the parent is heterozygous for the gene that affect this phenotype and that this gene is linked to this SNP. Thus, zygosity state can be heterozygous or homozygous, which is parallel for segregating or not segregating for the QTL. We rephrased "This test compares the polymorphism zygosity states (homo- or heterozygous genotype) to the QTL segregation status for individuals…"
Line 65: Please “introduce” the FABP4 gene. Full name? Function? Locality? Has this gene been previously identified to play a role in RFI?
Location and the full name, which also state the function, are now indicated better stating that references 11 and 12 previously identified it to play a role in RFI:
"… RFI studies indicating that BTA14 fatty-acid-binding protein 4 (FABP4) has a large effect on this trait [11,12]."
Line 84: What pipeline? I suggest “We followed a previously-described whole-genome sequencing (WGS) analysis pipeline (Ref)”
Done.
Line 91: Please be specific. What GATK pipeline? Aren’t there many?
We added "… workflow for germline short variant discovery."
Line 55/Line 91/etc.: Throughout, please check that all abbreviations are defined upon the first usage e.g. SNPs
Done.
Line 100: Please be specific. What commercial kit? And following the manufacturer’s protocol?
We rephrased "Blood samples were collected in EDTA vacutainers, and DNA was extracted with GenElute Blood Genomic DNA commercial kit (Sigma-Aldrich, St Louis, MO, USA) following the manufacturer’s protocol.
Line 101-102: This does not make sense to me. Is the 777K not an Illumina® genotyping panel? Was GeneSeek the service provider that performed the actual genotyping? Please be specific.
GeneSeek is Illumina's first certified service provider (Illumina CSPro) and is partnering with Illumina.
Line 102-103: I do not understand this step. Can you please rephrase and/or elaborate? Does it mean that you only extracted and utilized SNPs in the region on BTA14? If so, please specify the region (i.e. base pair range).
It is clearly stated that we only extracted and utilized BTA14 SNPs. The analyzed windows are described in details in the supplement. We added the ranges: "genetic markers along BTA14 were selected in the analyzed windows (positions 16,732,918 - 16,848,253 and 44,656,120 - 44,690,793, see Dataset, Supplementary Materials)."
**In the previous section (i.e. 2.2), you then also need to provide more details on the WGS data – before the pipeline was utilized, what was the targeted coverage, etc.? And on what platform was the WGS done?
WGS was performed with Illumina HiSeq X Ten (90 Gb), or with NovaSeq platforms and has been described in details in our previous Genes manuscript [13]. We now better refer to this previous work by rephrasing: "We followed a previously-described WGS analysis pipeline [13]. Briefly, raw reads of 19 bulls (ENA BioProject PRJEB59761, [13]) were mapped to the reference genome…".
Line 107-111: The imputation section needs to be stand-alone. It needs to be elaborated on – please specify the parameters used (even if the default was used), and the reference-versus-validation population used (if so), the underlying imputation methodology used within the software.
Section "2.3. Genotyping by imputation" does stand alone. We further elaborated "Findhap.options file included the following parameters iters 6, Xchrom 0, maxlen 17, minlen 17, steps 1, maxhap 50, hapout 1, genout 2, damout 0, listout -1 and errate 0.00. For PLINK imputation the following command line was used: 'p-link --noweb --tped file.tped --tfam file.tfam --cow --hap-window 17 --hap-phase'."
I would suggest restructuring the entire M&M. It is not explicitly stated whether all data types were available for the same animals (i.e. did each animal have WGS, 777K, and Sanger data available?)… I would start off with a section on each data source i.e. WGS data > SNP genotyping panel data > Sanger sequencing data, and then imputation. Also, how was the imputation accuracy quantified? And what is the “RFI” column in the supplementary material? Are they phenotypes – there is no mention thereof in terms of origin (i.e. service provider), units (are they EBVs or performance records), etc., and needs to be added to the M&M?
We feel that the current M&M structure that first described the animals and the phenotyping is better than starting with the WGS data. Please note that this article is only briefly describes how the data of previous works were obtained and direct the reader to the original publications for detailed information. The current work only adds the 777K beadchips and Sanger genotypes.
It is clearly stated "A total of 192 cows were genotyped on the BovineHD BeadChip", thus, all cows had this data. On the other hand, as clearly indicated in Results, Sanger sequencing was only applied to validate the genotyping accuracy of the examined methods including imputation e.g., "This sub-sample of 25 cows was further validated by Sanger sequencing…". To further clarify this we add to M&M: "Sanger sequencing was applied to a limited sample, in cases of conflicting genotypes between the examined genotyping methodologies, or to validate the imputed genotypes." In the supplementary material, we indicted that the “RFI” column is the phenotype. For the sample tested in the current work, these measurements were described in details by previous publications as indicated in section "2.1. Population and phenotypes".
Section 2.5 requires improvement in terms of detail. E.g. “Then, the segregation of the candidate alleles with the marker allele was examined.” This is very vague. There is no mention of software and/or the actual method/formula/train of thought that was used. Is this a verified way of doing this? E.g. “Since the population frequencies of the candidate alleles were lower…” – how were the frequencies calculated? Or were they given (if so, where)?
We revised the entire section accordingly:
2.5. Obtaining candidate variant around BTA14 RFI QTL
From the joint VCF file of 19 bulls with WGS data, all likely functional variants (i.e., missense, frameshift, stop gain/loss, and splice variants) and their genotype in each of the sires were obtained from +/- 1Mb flanking the marker with the lowest probability of accepting the null hypothesis of no effect. A total of 18 likely functional SNPs were retained for further analysis (data not shown). Assuming linkage disequilibrium between the BovineHD SNP associated with RFI and the hypothetical causative variant, both are expected to tend to co-segregate. Therefore, the numeric representation of their genotypes (0, 1, 2) would correlate. The numeric representation was as follows: homozygote for the reference allele (0 non-reference alleles), the heterozygote (1 non-reference allele) and the non-reference homozygote (2 non-reference alleles). We thus calculate the Pearson correlation coefficient between the vector of the likely functional SNPs genotypes, to the vector of the significant SNP marker genotype. These steps pointed out one SNP in the gene zinc-fingers and homeoboxes 2 (ZHX2) with a maximal Pearson correlation coefficient of 0.71.
Line 132-137: This belongs in the M&M section – please see my comments for Lines 107-111 above.
Indeed, as introducing sentences to Results, we briefly repeated some information about the methodology. We feel that this writing style is more reader friendly.
There is also no mention of/reference to Table 1? Are the SNPs in Table 1 known variants? This needs to be mentioned in the Introduction. Also, there are many vague column headings – what is “Minor”? As per your footnote, how was MAF calculated? This is not mentioned in the M&M…
Table 1 is referred to four times in the manuscript. We placed it a little ahead of the first time in section "3.1. Imputing FABP4 SNP genotypes". As indicated in this section, this table contains some known variants have been described in [12]. As described in Results, the other variants were characterized in this study. MAF calculation is strait forward as it can also be inferred from the haplotype frequencies (Tables 3 and 5).
Line 142-144: Not certain what this sentence means? This detail (e.g. about the gene of interest and the SNPs harbored within/around its genomic location) should be discussed in your introduction.
Indeed, as introducing sentences to this section of Results, we briefly repeated some information about this gene. We feel that this writing style is more reader friendly. We also now better present this gene in Introduction using its full name: "… RFI studies indicating that BTA14 fatty-acid-binding protein 4 (FABP4) has a large effect on this trait [11,12]."
Line 151-152: What were the “test and the reference samples”? This needs to be discussed in your M&M. Are they related somehow? How do you know they share haplotypes? Furthermore, how do you know the imputation was accurate? This is not a very big sample size.
In section 3.1, it is clearly indicated that the 19 sires were the reference "These sires served as the representative reference sample." Thus, all the rest (the cows) is the test population. We added in this section: To infer unavailable genotypes for these SNPs in the enlarged test sample of cows…" The issue of test and the reference is discussed in detail in Introduction. We know that they share haplotypes as both samples are Israeli Holstein and some of cows are daughters of the reference sires. As stated in the manuscript, we evaluated the accuracy of imputation by comparing it to the HRM and Taqman methods. When we encountered conflicts the genotypes were verified by Sanger sequencing. This enabled us to conclude that imputation was the most accurate- hence, the title of the manuscript.
Line 168: Who indicated this? This is part of the motivation for this study, so it needs to be very clear.
The indication is in the last sentence of the previous section (our result) "four haplotypes were encountered, and none of them had a significant effect on RFI". To better indicate this, we added " Following the indication (section 3.1)…"
Line 167-188: This section is very confusing. Some parts belong in the M&M, and others are discussions (not results). I would suggest a combined Results and Discussion… You also reference too many different tables – it gets confusing. Each table should be discussed separately.
In this section, there are only two tables that are closely related, Table 4 gives the SNPs in BTA14 interval of the ZHX2 locus and Table 5 that shows the haplotypes formed by these SNPs. Thus, essentially both tables complete each other and are discussed together. This also follows the presentation style for the FABP4 locus, in the previous section. Generally, our style of writing for each result section begins with briefly stating of the specific motivation and methodology, followed by describing the result and its significance. We are aware that other researchers prefer a strict approach of Results without any indication of why and how they are shown and interpreted. Yet, we feel that Results should stand alone and our style of writing is commonly used in current literature.
Line 184: “Capable of explaining 98.1%...”. How was this calculated (no mention in M&M)?
The sum of the numbers in the Frequency column in Table 5 is 0.98, hence the 98.1% number. To clarify this we rephrased: "Capable of explaining 98.1% of allele genotypes combinations (sum of all haplotypes frequencies, Table 5)…"
Line 196-199: This analysis was never mentioned in the M&M? What software was used to do the significance testing? If this was mentioned in the M&M, no footnote would be necessary.
This footnote is anyhow necessary to explain the terminology within this table header. To indicate that this study is not focusing on association analysis of RFI, we preferred not to include description of association tests in M&M. As indicated in this footnote the software used was PLINK. The last remark of this reviewer that asks "Is this an association study…?" further stresses the need for such policy.
Line 201-207: This needs to be described in the M&M. It should not be necessary to explain how things were done in your Results section – only results should be reported.
Generally, our style of writing for each result section begins with briefly stating of the specific motivation and methodology, followed by describing the result and its significance. We are aware that other researchers prefer a strict approach of Results without any indication of why and how they are shown and interpreted. Yet, we feel that Results should stand alone and our style of writing is commonly used in current literature.
Line 213: “For two individuals (3187 and 3330),…” Not necessary to mention animal IDs – the reader knows nothing about these animals – rather describe them in terms of sex or age.
The supplementary Dataset provides the reader with the information of all individuals identified by animal IDs including these two.
Line 216: How were these error rates calculated?
The calculation of these error rates is shown in Figure 1. E.g., it is obvious that in the HRM column there were 15 errors (cells with red background) of which two were excluded as problems in animal identity (red background in all methods), hence (15-2)/112=11.6%.
Figure 1a): It should not be necessary to include this (i.e. an Excel spreadsheet indicating genotypes per individual) – provide means or proportions of animals with correct versus incorrect (or differing) genotypes.
The short table in Figure 1a is actually the highlight of this study and therefore should not be hidden in an attached Excel spreadsheet.
Line 220: This is the first time the full name of the gene is mentioned…
We added full names for the genes in their first occurrences also in the body of the manuscript.
Figure 1’s title is too long. The majority of this should move to the text.
This title is only of 9 words: "Resolving genotyping discrepancies in bovine Fatty-acid-binding-protein 4 gene (FABP4)." The rest is the figure legend, which we believe should provide detailed information to allow this figure to stand alone.
Line 235-236: The regression analysis is never mentioned in the M&M.
We now mention this basic tool at the end of section 2.3: "Following imputation, the effects of the candidate alleles on RFI were assessed by linear regression, and coefficients of determination and regression were calculated using Excel (Microsoft Corporation, Redmond, WA, USA)."
Line 237: “In this figure,…” What figure? Please be specific.
We rephrased: "in Figure 2…"
Line 240-243: Please rephrase – this seems like an important result; however, it is not clear enough. I am also not sure what the second sentence is saying (the one referring to Figure 2b).
We clarified: "Moreover, TaqMan genotyping errors were capable of skewing the effect size by reversing its direction, as evident from the positive slope in Figure 2b compared to the negative slopes in Figure 2a and 2b."
Line 269-270: Yes, but how was imputation accuracy determined? And what was the composition of the reference population in terms of relatedness etc.? Were these high-impact bulls? If so, you need to provide some more details as this is one of the most important aspects of imputation. If the haplotype diversity is sufficiently captured, and with WGS information nonetheless, then the size of the reference data set should have less of an effect.
The accuracy of imputation is determined by Figure 1a. As stated in the manuscript, we evaluated the accuracy of imputation by comparing it to the HRM and Taqman methods. When we encountered conflicts the genotypes were verified by Sanger sequencing. This enabled us to conclude that imputation was the most accurate- hence, the title of the manuscript.
The impact of the bulls was described in the first section of results "We also performed whole genome sequencing (WGS) for 19 Israeli Holstein sires, representing major lineages of the Israeli herd [20] with no RFI evaluations." The relatedness matrix of these bulls introduced in that work [20].
Line 274: What is the 78%? What parameter? Is it a concordance rate? You need to be specific on what “similarity” means.
We refer to the results presented in [12]. There the sample size was 114, which means that 0.78*114=89 cows had identical genotypes for both mutations.
We clarified: "…(78% identical genotypes)…"
Line 286: How was relatedness quantified?
Relatedness was not quantified, but it is obvious as generally the effective size of the Holstein population is low (e.g., 39 in the USA - Weigel KA. Journal of Dairy Science. 2001, 84) thus taking 19 sires (38 chromosome sets) that are related to Israeli Holstein are likely to provide a suitable reference as was demonstrated in our manuscript. Further information regarding the relatedness matrix of these bulls introduced in a previous work [20].
Line 292: This implies that you used EMMAx in this study, however, it is not mentioned once in your M&M? Also, what PLINK functionality was used? Also association analysis? Not clear at all.
To prove our point that FABP4 effect on RFI was overestimated, we provide another BTA14 gene with a larger effect. We used association tests to locate this example but this is not the focus of our study. Moreover, we declare that "searching for the causative sequence variation that underlies RFI should await the development of statistical methods that would be able to handle additive and epistatic interactions". Therefore, we avoid describing the association testing as tools that convincingly point to causative variations. Yet, we added a description to the supplement and rephrased in section 3.2: "we reevaluated the association of BTA14 BovineHD SNPs with this trait using PLINK and EMMAX software (Supplementary Materials, data not shown)."
Line 294-296: In what species?
We added "In mice…"
Line 296: What is SIFT score? I am not following.
SIFT score is a normalized probability of observing the new amino acid at certain position. It ranges from 0 to 1. We refer to [27] for further explanation. Thousands of scientific publications refer to this score as a measure to asses the possible effects of mutations on the protein function. We rephrased: "… SIFT score of 0.03 (SIFT is an algorithm that provides a prediction of how likely nucleotide substitution is functional/damaging [27])."
Line 303-309: I am not sure if I am missing something but there is a lot of focus on HRM and TaqMan methodologies but these are not included in this study. The focus should be on the Sanger, SNP genotyping, and WGS data.
We took published results of HRM and TaqMan and reevaluated them using imputation, concluding that the latter is better. We also realized that using HRM and TaqMan led to erroneous conclusions affecting RFI research, which we rectified using the better genotyping. Thus, although we did not perform HRM and TaqMan these are included in our study by using data that had been produced previously.
There is not enough discussion on the relevance, impact, and possible benefits of your results. How can the results reported here be used to optimize future genotyping strategies? Is it a cost-benefit? The take-home message and purpose of the study are not coming across very well. How does this affect the feasibility of certain data types?
The take-home message is clearly stated in the next section of Conclusions (and the title of this manuscript):
"genotyping by imputation … produced superior results compared to other methods..." and "should also be regarded as a method of choice to validate the reliability of the genotypes"
Thus, other genotyping data if not validated may promote serious mistakes in scientific research as exemplified in the described case of RFI study.
Line 311: “Our case of association study…” Is this an association study…? I think you need to change your title as it is misleading considering the results reported. My perception was that imputation methodology is the main focus, however, that is not the case.
We rephrased: "Our case of a reevaluation of genotyping methodologies used to study dairy-cattle RFI demonstrated…"
Round 2
Reviewer 2 Report
All my questions were addressed. No further comments.